# Miniscrew-Assisted Rapid Palatal Expansion (MARPE): An Umbrella Review

**DOI:** 10.3390/jcm11051287

**Published:** 2022-02-26

**Authors:** Vanda Ventura, João Botelho, Vanessa Machado, Paulo Mascarenhas, François Durand Pereira, José João Mendes, Ana Sintra Delgado, Pedro Mariano Pereira

**Affiliations:** 1Orthodontic Department, Instituto Universitário Egas Moniz, 2829-511 Almada, Portugal; fdupereira@netcabo.pt (F.D.P.); adelgado@egasmoniz.edu.pt (A.S.D.); pmarianop@egasmoniz.edu.pt (P.M.P.); 2Clinical Research Unit (CRU), Centro de Investigação Interdisciplinar Egas Moniz (CiiEM), Egas Moniz–Cooperativa de Ensino Superior CRL, 2829-511 Almada, Portugal; jbotelho@egasmoniz.edu.pt (J.B.); vmachado@egasmoniz.edu.pt (V.M.); pmascarenhas@egasmoniz.edu.pt (P.M.); jmendes@egasmoniz.edu.pt (J.J.M.); 3Evidence-Based Hub, Clinical Research Unit, Centro de Investigação Interdisciplinar Egas Moniz, 2829-511 Almada, Portugal

**Keywords:** miniscrew-assisted rapid palatal expansion, MARPE, maxillary expansion, umbrella review, orthodontics, orthopedics

## Abstract

In postpubertal patients, maxillary transverse discrepancy is a common condition often requiring surgical approaches. To overcome the excess morbidity and discomfort, maxillary expansion through miniscrew-assisted rapid palatal expansion (MARPE) was proposed and studied in the last few years. This umbrella review aims to critically appraise the quality of evidence and the main clinical outcomes of available systematic reviews (SRs) on MARPE. An extensive search was carried out in five electronic databases (PubMed-Medline, Cochrane Database of SRs, Scielo, Web of Science, and LILACS) until December 2021. The methodological quality was appraised using the A Measurement Tool to Assess SRs criteria 2 (AMSTAR2). The primary outcome was the methodological quality of SRs. Overall, four SRs were included and analyzed, one of high methodological quality, one of low and two of critically low. Despite the verified methodological constraints, MARPE seems to present significant clinical changes when compared to conventional RPE, SARPE or controls and less adverse clinical outcomes. The quality of evidence produced by the available SRs was not favorable. Future high standard SRs and well-designed clinical trials are warranted to better clarify the clinical protocols and outcomes success of MARPE.

## 1. Introduction

Maxillary transverse discrepancy is a common condition in orthodontic subjects affecting adult population [1,2,3]. Usually, this discrepancy results in a unilateral or bilateral posterior crossbite phenotype [4,5] and often requires intervention with the maxillary disjunction, being a predictable procedure in prepubertal patients [6,7]. At this stage, rapid maxillary expansion (RME) devices (known as rapid palatal expansion [RPE]) are routinely used [8] to separate the midpalatal suture, followed by orthopedic maxillary expansion [2,6,7,9]. In postpubertal patients (young adults and adults), the application of RME techniques is controversial because median palatal suture is often maturated and is clinically difficult to open. Consequently, these techniques may result in undesirable side effects, namely dentoalveolar compensation [8], dental and periodontal undesired effects [10,11,12] in involved teeth rather than true skeletal expansion. Thus, surgically assisted maxillary palatal expansion (SARPE) is an orthodontic treatment that aims to manage the maxillary transversal discrepancy through opening circummaxillary sutures in postpubertal patients [13,14,15,16,17,18,19,20], although the high costs and increased post-surgical morbidity [21]. Areas of greatest resistance to maxillary expansion are the midpalatal suture, the zygomatic apophyses, pterygomaxillary sutures (pterygopalatine). Histological studies have shown that the midpalatal suture starts closing in pre-puberty and reaches a high level of interdigitation, leading to a greater closure puberty [2,6].

A rigid element that transfers the expansion force directly to the basal bone may allow a disjunction in post-pubertal patients [11,22,23,24]. To achieve such therapeutic purpose, miniscrew-assisted rapid palatal expansion (MARPE) [22,23,24] devices have been developed and studied in recent years [14,22,23,24,25,26,27,28]. When compared to conventional RPE techniques, MARPE reduces the risk of dentoalveolar compensations and undesirable effects in post-pubertal patients [11,12,29]. When compared to SARPE, MARPE is a simpler technique and has lower impact on patient-reported outcomes and lower costs [8,11], and shifting from a complete analogic protocol to the incorporation of a digital workflow has been proved possible [30]. Furthermore, recent systematic reviews have been produced on MARPE, although their methodological quality and evidence remain to be critically appraised [10,11,31,32].

Therefore, the present umbrella review aimed to critically appraise the available systematic reviews (SRs) on MARPE, with a particular two-fold focus: (1) to assess the quality of evidence; and (2) to summarize its clinical outcomes.

## 2. Materials and Methods

The protocol for this umbrella review was defined *a priori* by all authors and was performed following the Preferred Reporting Items for SRs and Meta-Analyses (PRISMA) guidelines [33], expanded with the guideline for SRs of SRs [34]. We aimed to answer the following main research question: “What is the quality of evidence of SRs on MARPE?”

### 2.1. Study Selection

For this umbrella review, five electronic databases (PubMed-Medline, Cochrane Database of SRs, Scielo, Web of Science, and LILACS) were searched from the earliest data available until December 2021. We merged keywords and subject headings in accordance with the thesaurus of each database: (“microimplant-assisted rapid palatal expansion” OR “miniscrew-assisted rapid palatal expander” OR “micro-implant-assisted rapid palatal expander” OR “mini-implant-assisted rapid palatal expander” OR MARPE OR “Palatal Expansion Technique” [MeSH]) AND (systematic review). Gray literature was searched through the OpenGrey portal (http://www.opengrey.eu, accessed on December 2021). Additional relevant literature was included after a manual search of the reference lists of the final included articles. 

Electronic search database was carried out by two independently authors (V.V. and J.B.), and the final decision for inclusion was made according to the following criteria: (1) SRs with or without meta-analysis; (2) human trials; (3) assessing clinical characteristics of MARPE. There were no restrictions regarding year of publication nor language. 

### 2.2. Information Sources Search

A predefined table was used to extract necessary data from each eligible SR, including the first author’s name, publication year, type of included studies, number of cases (control and interventional group, if applicable), interventions, outcomes, tool used to assess the quality of studies, main results and main conclusion and funding. From each eligible SRs, two researchers (V.V. and J.B.) independently extracted information and all disagreements were resolved through discussion with a third reviewer (V.M.). Outcomes were classified as: success rate; side effects; and impact on airway and nasal breathing. 

### 2.3. Risk of Bias Assessment

Risk of bias of included SR was independently assessed by two calibrated authors (V.V. and J.B.) using the MeaSurement Tool to Assess SRs (AMSTAR 2) [34]. According to this tool, SRs are categorized as: High (‘Zero or one non-critical weakness’); Moderate (‘More than one non-critical weakness’); Low (‘One critical flaw with or without non-critical weaknesses’); and Critically Low (‘More than one critical flaw with or without non-critical weaknesses’). The final quality rate was obtained for each study.

## 3. Results

### 3.1. Study Selection

The database search resulted in 376 records. After excluding 208 duplicates (Figure 1), we assessed the title and abstract of the remaining entries, resulting in seven potentially eligible full-text studies. As a result, three studies were excluded (Table 1), resulting a final number of four systematic reviews. The full-text screening reliability among examiners was excellent (kappa score = 1.00).

### 3.2. Study Characteristics

Overall, two SRs [10,31] and two SRs with meta-analysis [11,32] were included (Table 2). Multiple sub-topics were investigated, such as clinical success rate, oral adverse effects of MARPE, skeletal and dental transverse expansion rate, airway changes (volume, nasal function and nasal airway resistance).

### 3.3. Methodological Quality Assessment

The reliability of RoB screening among examiners was categorized as excellent (kappa score = 0.91; 95% confidence interval: 0.89–0.92).

The AMSTAR2 Criteria was not satisfied entirely by any of the included systematic reviews (Table 3). Overall, one was rated as ‘high quality’ [33], one as ‘low quality’ [11] and two were assessed as ‘critically low quality’ [10,32]. The major problems were found on the: (i) reporting on the sources of funding for the studies included in the review (*n* = 4, 100%); (ii) explaining their selection literature search strategy (*n* = 3, 75%); (iii) providing the list of excluded studies and exclusions justified; (iv) and appraising the existence of publication bias. Publication bias was not deemed possible to compute in one study given the limited number of included studies (<10) [11], while in another study meta-analysis was not carried out [33].

### 3.4. Synthesis of Results

#### 3.4.1. Success Rate

The mean success rate was estimated at 92.5% (between 80.7% and 100%) [11]. 

#### 3.4.2. Skeletal Transverse Maxillary Expansion

MARPE, in what skeletal transverse maxillary expansion concerns, skeletal width increased by 2.33 mm (95% CI: 1.11–4.5 mm), while immediate post-expansion rate 35.6% (95% CI: 25–61%) [11]. 

When compared MARPE to SARPE and RPE, SARPE was estimated to present higher skeletal expansion mean 3.3 mm [31], (21.5% to 46.3%) [31], and RPE showed less skeletal expansion (40% to 55%) [31].

#### 3.4.3. Dental Transverse Maxillary Expansion

One study addressed dental transverse maxillary expansion and concluded an average increase of intermolar width of 6.55 mm (ranging from 5.4 mm to 8.32 mm), an average increase of intercanine width (ICW) between 2.86 mm and 5.83 mm, and an average increase of interpremolar width (IPW) between 5.33 mm and 6.09 mm [11].

#### 3.4.4. Duration of Expansion

The mean number of days of expansion ranged between 20 to 126 until the necessary amount of expansion was achieved, through different expansion protocols [11]. Specifically, the duration of rapid expansion protocols ranged between 20 and 35 days [11].

#### 3.4.5. Oral Adverse Effects

Although MARPE is a short-term treatment, dental tipping was a highly reported dental effect, despite important the methodological differences among studies [11]. Overall, dental tipping of the first molar was statistically significant and ranged from 2.07° to 8.01°, with similar results to RPE and SARPE [11].

In addition, there is limited evidence on the impact of MARPE on the periodontium (specifically, bone thickness and marginal bone level), whose changes are considered clinically insignificant and comparable to RPE [10,11]. To some extent, MARPE may produce less loss of buccal alveolar bone thickness and marginal bone level in the region of first premolars than the conventional RPE [10]. 

Concerning soft tissue effects, MARPE was reported to cause short-term impact on nasal soft tissues, the majority of which showed significant positional changes [11]. The nose tends to widen and move forward and downward and the post-treatment nasal volume exhibits an increase relative to the initial volume [11].

#### 3.4.6. Airway Volume, Nasal Function and Nasal Airway Resistance

Evidence on this thematic is contradictory. Based on studies with low to moderate evidence, Abu Arqub et al. [32] reported poor association between nasal function and airway volume, and MARPE does not lead to a significant change in the airway volume compared to RPE and controls in young children and adolescents between 10- and 17-years-old.

Furthermore, studies reported short-term significant change in the muscle strength [32], with decrease of nasal and airway resistance and airflow of MARPE over conventional RPE [31,32]. Similarly, MARPE was found to positively impact nasal breathing [31].

#### 3.4.7. Subjective Patient-Reported Changes in Nasal Breathing

MARPE was significantly associated to self-perceived improvement in breathing after maxillary expansion, but not sufficiently maintained until 18 months of follow-up [31].

## 4. Discussion

### 4.1. Summary of the Main Results

From the included SRs, the quality of evidence is unfavorable, with only one SR being of high quality. Despite the verified methodological constraints, MARPE seems to present significant clinical changes when compared to conventional RPE, SARPE or controls and less adverse clinical outcomes.

Our findings may have relevant implications in future investigations to reach precise and evidence-based results based on widely accepted guidelines. On the one hand, the scientific prematurity of studies of MARPE is mirrored in the still low number of overall patients included in the SRs. This fact ultimately decreases the scientific certainty and evidence robustness. On the other hand, the majority of SRs failed to comply with critical domains according to the AMSTAR2 tool and the PRISMA guideline. Therefore, future systematic reviews on MARPE require additional effort on: reporting on the sources of funding for the studies included in the review; explaining their selection literature search strategy; providing the list of excluded studies and exclusions justified; and, appraising the existence of publication bias.

### 4.2. Quality of the Evidence and Potential Biases in the Review Process

Although the different systematic reviews are mostly based on studies with high and moderate (moderate to serious) risk of bias, all studies report that MARPE has high clinical success rate and significant maxillary skeletal expansion. These results are in agreement with the values found in the SARPE studies. Although these values are statistically different from SARPE, they are clinically insignificant.

As in SARPE and RPE, some secondary effects were found: dentoalveolar, periodontal and soft tissue effects. These effects were smaller when compared to RPE and larger when compared to SARPE. It was also confirmed MARPE also influences the upper airways, increasing/improving nasal breathing (nasal breathing/nasal air flow), decreasing nasal resistance and decreasing NOSE (nasal obstruction scale) score at least in a short-term, which is not clear if it remains in the long-term, since the studies have different and short follow-up.

Yet, several shortcomings have to be borne in mind. Particularly, we emphasize the level of risk of bias, the lack of high-quality studies, the low number of systematic reviews available, the relatively low number of articles on the topic, as well as the different ages of the patients (some even in a growing process). Furthermore, there was substantial methodological variability, such as the type of devices used, the device support locations, the expansion protocol, the numbers of miniscrews used, and the measurements taken.

MARPE is thus an alternative to SARPE, for post-pubertal patients, with statistically lower but clinically insignificant results compared to SARPE, which has higher costs with hospitalization and morbidity. Further studies should be done to confirm the data found.

In this umbrella review, some strengths and limitations can be found. Overall, we carried out a transparent and evidence-based strategy of search and appraise. Yet, readers must bear in mind that interpretation of such results derive from conclusions lean on the interpretation of the original SRs. This umbrella review was only able to include four SRs, which can be explained by the relatively novelty of this approach and clinical studies on this respect, nevertheless we conclude the methodological quality of such SRs in not favorable and therefore our conclusions may serve to alert the scientific community to the need for better future SRs. 

The number of available clinical trials is also a requirement, that indeed will advance medical knowledge on MARPE and improve patient care. A possible challenge will be to focus on dental patient-reported outcomes (dPROs) (such as, perceived quality of life, pain experience or satisfaction with the results), that encompasses both minors and adults. For each group, different sets of dPROs are available and may be interesting in future research [38].

## 5. Conclusions

The SRs performed on MARPE revealed an insufficient quality of evidence. In the future, SRs are required to comply with comprehensive guideline and clinical trials are warranted to better elucidate the most effective MARPE clinical protocols and its success rates.

## Figures and Tables

**Figure 1 jcm-11-01287-f001:**
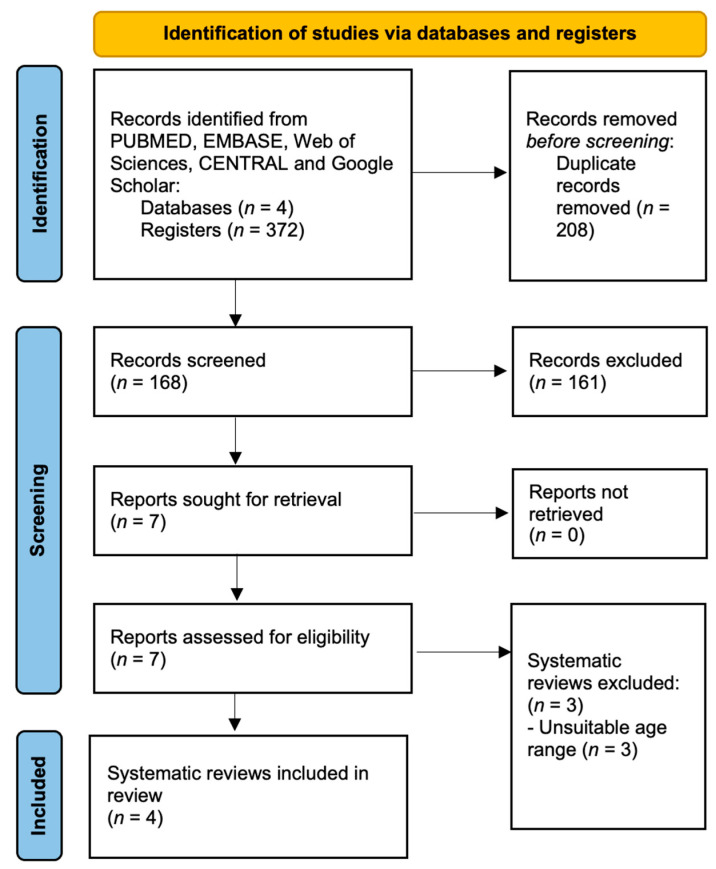
PRISMA flowchart.

**Table 1 jcm-11-01287-t001:** List of studies excluded with respective reason.

Reference	Reason for Exclusion
Krüsi et al. (2019) [35]	Unsuitable age range
Khosravi et al. (2019) [36]	Unsuitable age range
Hassan et al. (2021) [37]	Unsuitable age range

**Table 2 jcm-11-01287-t002:** Overview of the included studies.

Authors (Year)	Search Period	N and Type of Studies	Patients	Outcome	Quality Assessment Tool	Main Results (ES [95%CI]) (I2)	Conclusions	Funding
Abu Arqub et al. (2021) [32]	Up to June 2020	2 RCTs and 1 prospective NRSI	121	Airway size, volume, and function	ROB and ROBINS-I	No MA	The short-term airway volumetric changes secondary to MARPE were not significant. The influence of MARPE appliances on breathing is still not clear.	None
Calvo-Henriquez et al. (2021) [31]	Up to April 2020	10 case series studies	257	Subjective measures (visual analogue scales, questionnaires assessing sinonasal symptoms or any other quantitative) and objective measurements (rhinomanometry, rhinohigrometry, fluid dynamics simulation, or peak nasal flow, among others).	Checklist from the NIH and clinical excellence	Nasal resistance—SMD (0.27 [0.15, 0.39]) (5.0%) NOSE score—SMD (40.08 [36.28, 43.89]) (25.0%)	Available evidence is too limited to suggest maxillary expansion as a primary treatment option to target nasal breathing	None
Kapetanović et al. (2021) [11]	Up to 20th November 2020	2 prospective and 6 retrospective NRSI	259	Success rate, skeletal width and dental intermolar width	ROB and ROBINS-I	No MA	MARPE is associated with a high success rate in skeletal and dental maxillary expansion. MARPE can induce dental and periodontal side effects and affect peri-oral soft tissues	None
Copello et al. (2020) [10]	Up to January 2020	3 RCTs and 1 retrospective NRSI	155	Buccal alveolar bone thickness and/or marginal bone level (bone dehiscence)	ROB ROBINS-I	SMD (0.55 [0.29, 0.80]) (40.0%)	Limited evidence suggests that MARPE could decrease the loss of the buccal alveolar bone when compared to conventional RPE	Research grant

ES—Effect Size; I^2^—Heterogeneity (measured in %); NIH—National Institute for Health; NRSI—Non-randomized studies of intervention; RCTs—Randomized Clinical Trials; ROB—Risk of bias by Cochrane; ROBINS-I—Risk of Bias in Non-Randomized Studies—of Interventions; SMD—Standardized Mean Difference.

**Table 3 jcm-11-01287-t003:** Methodological quality of the included systematic reviews.

Authors (Year)	1	2	3	4	5	6	7	8	9	10	11	12	13	14	15	16	Quality
Abu Arqub et al. (2021) [32]	Y	Y	Y	PY	Y	Y	Y	Y	Y/Y	N	0	0	Y	Y	0	Y	High
Calvo-Henriquez et al. (2021) [31]	Y	Y	N	N	Y	Y	N	Y	Y	N	Y/Y	Y	N	Y	Y	Y	Critically Low
Kapetanović et al. (2021) [11]	Y	Y	N	Y	Y	Y	Y	PY	Y/Y	N	Y/Y	Y	Y	Y	0	Y	Low
Copello et al. (2020) [10]	Y	Y	N	Y	Y	Y	N	Y	Y/Y	N	Y/Y	Y	Y	Y	N	Y	Critically Low

0—No meta-analysis conducted, N—No, Y—Yes, PY—Partial Yes. 1. Research questions and inclusion criteria? 2. Review methods established a priori? 3. Explanation of their selection literature search strategy? 4. Did the review authors use a comprehensive literature search strategy? 5. Study selection performed in duplicate? 6. Data selection performed in duplicate? 7. List of excluded studies and exclusions justified? 8. Description of the included studies in adequate detail? 9. Satisfactory technique for assessing the risk of bias (RoB)? 10. Report on the sources of funding for the studies included in the review? 11. If meta-analysis was performed, did the review authors use appropriate methods for statistical combination of results? 12. If meta-analysis was performed, did the review authors assess the potential impact of RoB? 13. RoB accounted when interpreting/discussing the results of the review? 14. Did the review authors provide a satisfactory explanation for, and discussion of, any heterogeneity observed in the results of the review? 15. If they performed quantitative synthesis, was publication bias performed? 16. Did the review authors report any potential sources of conflict of interest, including funding sources?

## Data Availability

Not applicable.

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
