# Peer review of "Miniscrew-Assisted Rapid Palatal Expansion (MARPE): An Umbrella Review"

_jcm, 2022, doi:10.3390/jcm11051287_

Round 1

Reviewer 1 Report

This umbrella review describes an interesting topic in the current orthodontic literature "Miniscrew-Assisted Rapid Palatal Expansion". The review is conducted well but there are a few concerns that need to be addressed. The authors should describe the risk of bias in more detail. The discussion can be expanded.  

Author Response

We are delighted to resubmit our revised manuscript titled "Miniscrew-Assisted Rapid Palatal Expansion (MARPE): An Umbrella Review" (Manuscript ID jcm-1580364).

All comments were considered and a point-by-point response and description of changes made are detailed below. In addition, please find appended a marked-up version of the manuscript with the changes to the text in red font and a clean version of the updated manuscript.

We hope our responses are now suitable for publication in the Journal of Clinical Medicine.

REVIEWER 1

This umbrella review describes an interesting topic in the current orthodontic literature "Miniscrew-Assisted Rapid Palatal Expansion". The review is conducted well but there are a few concerns that need to be addressed. The authors should describe the risk of bias in more detail. The discussion can be expanded.  

Answer: we greatly appreciate your time reviewing our manuscript.

To describe in more detail the risk of bias, we added the following sentence:

“Publication bias was not deemed possible to compute in one study given the limited number of included studies (<10) [11], while in another study meta-analysis was not carried out [33].”.

To expand the Discussion we added the following two sentences:

“Therefore, future systematic reviews on MARPE require additional effort on: reporting on the sources of funding for the studies included in the review; explaining their selection literature search strategy; providing the list of excluded studies and exclusions justified; and, appraising the existence of publication bias.”

as well

“The number of available clinical trials is also a requirement, that indeed will advance medical knowledge on MARPE and improve patient care. A possible challenge will be to focus on dental patient-reported outcomes (dPROs) (such as, perceived quality of life, pain experience or satisfaction with the results), that encompasses both minors and adults. For each group, different sets of dPROs are available and may be interesting in future research [39].”

Reviewer 2 Report

Thank you very much for the opportunity to review an exceptionally thoughtful and correct work.

The work is done correctly, both substantively and technically. 

My only comment is: 
in table 2, the works could be arranged chronologically.

Author Response

We are delighted to resubmit our revised manuscript titled "Miniscrew-Assisted Rapid Palatal Expansion (MARPE): An Umbrella Review" (Manuscript ID jcm-1580364).

All comments were considered and a point-by-point response and description of changes made are detailed below. In addition, please find appended a marked-up version of the manuscript with the changes to the text in red font and a clean version of the updated manuscript.

We hope our responses are now suitable for publication in the Journal of Clinical Medicine.

REVIEWER 2

Thank you very much for the opportunity to review an exceptionally thoughtful and correct work.

The work is done correctly, both substantively and technically. 

My only comment is: 
in table 2, the works could be arranged chronologically.

Answer: we greatly appreciate your time reviewing our manuscript. We have rearranged table 2, chronologically, as well as in Table 3.

Reviewer 3 Report

I read your article “Miniscrew-Assisted Rapid Palatal Expansion (MARPE): An Umbrella Review” carefully.

The paper looks very well built, even if the topic isn’t current and particularly interesting.

I think the introduction should be improve.

I invite you to consider the article

https://doi.org/10.3390/bioengineering9010031

I believe there are minor concerns.

Table 1 is unnecessary.

Overall, the article is well written. Check underlined text on page 6 line 178.

I look forward to seeing this article corrected accordingly to provide a final decision.

Author Response

We are delighted to resubmit our revised manuscript titled "Miniscrew-Assisted Rapid Palatal Expansion (MARPE): An Umbrella Review" (Manuscript ID jcm-1580364).

All comments were considered and a point-by-point response and description of changes made are detailed below. In addition, please find appended a marked-up version of the manuscript with the changes to the text in red font and a clean version of the updated manuscript.

We hope our responses are now suitable for publication in the Journal of Clinical Medicine.

REVIEWER 3

I read your article “Miniscrew-Assisted Rapid Palatal Expansion (MARPE): An Umbrella Review” carefully.

The paper looks very well built, even if the topic isn’t current and particularly interesting.

Answer: we greatly appreciate your time reviewing our manuscript. All your commentaries were considered and thoughtfully resolved.

I think the introduction should be improve.

I invite you to consider the article

https://doi.org/10.3390/bioengineering9010031

Answer: We have improved the introduction as per your recommendation and considering the article https://doi.org/10.3390/bioengineering9010031. The new added sentence reads “(…) and shifting from a complete analogic protocol to the incorporation of a digital workflow has been proved possible [31].”

I believe there are minor concerns.

Table 1 is unnecessary.

Answer: Although its usability is debatable, we have followed the AMSTAR2 and PRISMA guidelines, that demand placing the list with detailed reasons for exclusion.

Overall, the article is well written. Check underlined text on page 6 line 178.

Answer: We have removed the underlined format, accordingly.